# Managing Native Hip Protrusio: Simplified Classification and Surgical Recommendations

Lidia Ani [ID], Zachary Radford and Lee E. Rubin *[ID]

Department of Orthopaedics & Rehabilitation, Yale-New Haven Hospital & Yale University School of Medicine, 47 College Street, New Haven, CT 06510, USA
* Correspondence: lee.rubin@yale.edu; Tel.: +1-203-785-2579; Fax: +1-203-785-7132

**Abstract:** Protrusio acetabuli is a unique osteological condition that has been long described in the literature and is known to potentially increase both the surgical complexity and the risk of complications when performing total hip arthroplasty. Although grading systems for native hip protrusio have been described in the past, there has not yet been a widely adopted classification system that categorizes the condition into separate classes in order to guide management. We propose a novel classification system with the goal of simplifying and standardizing the management of protrusio acetabuli in the context of modern total hip arthroplasty. This classification system describes protrusio based on the relationship of the femoral head to the ilioischial and iliopectineal lines, allowing for a more reproducible and consistent categorization of pathology. We also discuss general recommendations and technical pearls for total hip arthroplasty in the protrusio patient population, including the use of strategic soft tissue releases, fluoroscopy, navigation, bone graft, and augments.

**Keywords:** protrusio; hip; acetabulum; total hip arthroplasty (THA); classification; direct anterior approach (DAA)





## 1. Introduction

Protrusio acetabuli is a condition characterized by protrusion of the acetabulum and femoral head into the pelvis [1]. It is defined by a lateral center edge angle greater than 40 degrees, in which the lateral center edge angle, or angle of Wiberg, is described as the angle formed by the intersection of a vertical line through the center of the femoral head and a line from the lateral edge of the acetabulum through the center of the femoral head [2]. Diagnosis of protrusio acetabuli can also be made based on protrusion of the medial wall of the acetabulum past the ilioischial line, or Kohler line, by a minimum of 3 mm in men or 6 mm in women [3,4].

Otto was the first to describe the condition in 1824, leading to the term Otto's disease, or Otto pelvis [5]. The translation of his description details the acetabulum as protruding into the pelvis "like half an orange", and notes that "the acetabulum protrudes so far into the pelvis that it contains not only the head but also the normally long neck of the femur, and the upper acetabular margin articulates with the greater trochanter" [1]. In 1929, Verrall contributed to the descriptions of protrusio with the name arthrokatadysis, or "subsidence of a joint" [6]. Verrall wrote that he believed the condition to be one "of some interest and rarity" and purported that the primary condition is the "sinking-in", which according to the author might be a representation of a "localized osteomalacia" [6]. Since then, several studies have been published regarding the diagnosis and management of protrusio [1,2,7,8]. Idiopathic, or primary, protrusio is diagnosed when no definitive causative factor is found for the condition and is therefore considered to be a diagnosis of exclusion [2]. Some even suggest that many cases of patients diagnosed as having primary protrusio acetabuli are in fact patients in which a subtle metabolic abnormality has been overlooked and is the true cause of the condition [2]. The pathophysiology of idiopathic protrusio is unclear, however,

some suggest that it may be related to persistent medial bulging of the acetabulum during development [2,9]. This bulge is thought to be normal in early childhood due to a lack of fusion of the triradiate cartilage but is expected to correct with age [1,2,10]. Failure of correction of this typically reversible protrusion is believed by some to be the cause of protrusio in the adult patient [2].

Multiple secondary etiologies of protrusio have been described, including infectious, neoplastic, inflammatory, metabolic, traumatic, and genetic causes [2,7,11]. Many early described cases of protrusio were thought to be secondary to septic arthritis that had gone untreated, especially in instances of gonococcal and tuberculous infections [2]. Reports have also been made of acetabular deficiency secondary to metastatic disease [2]. Additionally, certain genetic diseases have been found to have an association with protrusio acetabuli. Osteogenesis imperfecta, for example, is a genetic disorder that has a prevalence of protrusio reported to be as high as 54% [12–14]. Patients with rheumatoid arthritis as well had historically been reported to have high rates of protrusio acetabuli, with a prevalence of approximately 14% in this patient population during the era when steroids remained a mainstay of treatment [15,16]. The previous literature on ankylosing spondylitis reported almost a 50% prevalence of protrusio in the studied patient population [17], in stark contrast to the generally estimated incidence value for protrusio acetabuli in patients with osteoarthritis, which is reported to be around 5% [2]. Inflammatory etiologies of secondary protrusio acetabuli are thought to contribute to the condition by inflammation-related destruction of the bone in the vicinity of the hip joint, allowing migration of the hip joint along the direction of the joint reaction vector [2]. It has been suggested that after the joint reaction vector migrates medial to the ilioischial line, the rate of progression of the condition subsequently increases [2]. It is also important to recognize that in etiologies associated with low bone density, including osteogenesis imperfecta and rheumatoid arthritis, the progression of protrusio acetabuli may be hastened by osteoporosis-related insufficiency fracture of the medial wall of the acetabulum [15,18].

Given that total hip arthroplasty (THA) is in many cases the recommended treatment for adult patients with protrusio acetabuli and end-stage arthritis [19–21], determining the ideal strategy for protrusio management in THA is crucial. Grading of protrusio is the first step in this process, as assessment of deformity severity is necessary to select a management approach.

A few classification systems for grading of protrusio acetabuli have been described in the distant past. In 1978, Sotelo-Garza and Charnley separated protrusio into mild, moderate, and severe groups based on the distance between the acetabulum and the ilioischial line (Table 1) [22]. They classified mild protrusio as having this distance measure 1–5 mm, moderate protrusio as a measurement of 5–15 mm, and severe protrusio as measurements more than 15 mm [22]. They noted that the majority of hips analyzed were between Grade I (mild protrusio) and Grade II (moderate protrusio), with only 9.5% of hips in their study being classified as Grade III (severe protrusio [22]. Ranawat and Zahn were the next to contribute in 1986 when they suggested treatment patterns based on the degree of protrusio, attempting to provide a scaffolding on which surgeons could plan their operative techniques for fixation [23]. This classification rested on millimeters of displacement which correlated to the use of autograft, additional fixation, or neither [23]. In this classification system, the authors proposed that normalization of the center of rotation of the hip was the recommended goal in the management of protrusio acetabuli [23]. They suggested that this normalization could be completed by using autogenous bone graft with or without a threaded socket in patients with protrusio of more than 5 mm with an intact (though potentially thin) medial wall; and in patients with a grossly deficient medial wall, they advised reconstruction of the medial wall with bone graft in addition to a protrusio ring as well as any other fixation device, such as a threaded socket [23]. The addition of reinforcement in the form of mesh bone graft obtained from the femoral head or allograft was also recommended as an option by the authors [23]. In 1987, Hirst et al. published a similar system in which protrusio was also graded based on the distance between the

acetabular line and the ilioischial line (Table 2) [24]. The Hirst system factors gender into the classification and has a minimum distance required for protrusio diagnosis of 3 mm in men and 6 mm in women, rather than the 1 mm described in the Sotelo-Garza/Charnley study [22,24].

**Table 1.** Sotelo-Garza/Charnley classification for grading of protrusio acetabuli.

| Grade | Distance between Acetabulum and Ilioischial Line |
|:---:|:---:|
| I | 1–5 mm (mild protrusio) |
| II | 6–15 mm (moderate protrusio) |
| III | 16+ mm (severe protrusio) |

**Table 2.** Hirst classification for grading of protrusio acetabuli.

| Grade | Distance between Acetabular and Ilioischial Lines (Men) | Distance between Acetabular and Ilioischial Lines (Women) |
|:---:|:---:|:---:|
| I | 3–8 mm | 6–11 mm |
| II | 8–13 mm | 12–17 mm |
| III | Over 13 mm with fragmentation | Over 17 mm with fragmentation |

With regard to the application of the classification system to clinical practice, Hirst et al. commented that in moderate and severe cases of protrusio acetabuli, the use of a cemented acetabular component was likely to lead to construct breakdown due to poor containment provided by the compromised medial wall, and therefore was not advised [24].

Although some indication of the practical application of these classification systems exists, these "exact-distance" based systems are inherently limited in their clinical utility given the high variability of measurements that can be obtained for a single patient based on inconsistencies in image calibration that may exist from one image to the next. They are also limited and difficult to standardize from one patient to the next at a single center, as well as due to technique-related variability that exists from one institution to the next. As a result, classification systems with radiographic class delineations based on finite distances and exact distance measurements have limited consistency and reproducibility [25].

Classification systems have also been described for grading acetabular bone loss in the setting of revision total hip arthroplasty [26]. Some of the more commonly used classification systems for this indication include the Paprosky classification [27,28] and the American Academy of Orthopaedic Surgeons classification [29].

The Paprosky classification was developed in an attempt to systematically describe bone loss patterns in failed acetabuli [27]. The authors noted that bony destruction in failed acetabuli tended to progress in an orderly manner along a stepwise cascade. Proposed treatment options for each of the described defect types were also put forth by the authors [27]. They suggested that strict adherence to their classification system with observance of the recommended reconstruction approaches for each defect type could be expected to result in predictably acceptable results when revision acetabular surgery is performed [27].

In the Paprosky classification system, defects are categorized based on whether an intact acetabular rim is present or absent, as well as whether the remaining bone stock would be able to provide rigid support for an acetabular component [27]. The defects are classified into types that are defined by whether the bone stock that is remaining would be able to completely support, partially support, or not at all support an implanted component [27]. The defects can then be further divided into subtypes that relate to the specific fixation method that can be used for the management of the bony defect [27].

The American Academy of Orthopaedic Surgeons classification system is descriptive in nature, and unlike the Paprosky classification system, does not detail the specific reconstructive options that are available for a particular type of defect [29]. It also does

not directly describe the magnitude of the defect in question [29]. In both the Paprosky classification system and the American Academy of Orthopaedic Surgeons classification system, the definitive determination of the type of bone loss is performed intraoperatively, although it has been suggested that the defect can often also be classified using radiographs obtained prior to surgery [29]. Classifications systems such as those described so far have variably reported levels of intra- and interobserver reliability [28] and have to this point primarily been used to describe periprosthetic protrusion, typically associated with osteolysis surrounding failed hip replacement implants, rather than native hip protrusio.

There is a clear dearth of modern literature regarding a classification system for native hip protrusio acetabuli that can facilitate an organized approach to the management of this condition. All of the cited classification systems above were presented in 1987 or earlier [26–29], meaning that all are at least 35 years old at this time in 2022. As a result, radiologists and surgeons aiming to classify these deformities need a single, reproducible system for diagnosis. Moreover, treatment recommendations using modern techniques and technologies that are matched to each grade of deformity are needed and would be of immediate benefit to the treating surgeon.

In this paper, a novel classification system for protrusio acetabuli is proposed that we hypothesize will help guide management using modern hip reconstruction techniques and technologies. We also describe general recommendations and technical pearls for surgeons planning to perform total hip arthroplasty in patients with protrusio acetabuli.

## 2. Materials and Methods

Literature review was conducted to identify articles discussing protrusio acetabuli and complex total hip arthroplasty. Articles were assessed for relevance to the manuscript topic and those relating to the objective of the manuscript were included in the bibliography. The classification system for acetabular protrusio described in this manuscript was developed by the senior author. Surgical recommendations and technical pearls for managing acetabular protrusio were obtained from a combination of the senior author's personal experience and suggestions previously described in the literature cited in the bibliography. The protrusio cases highlighted in this manuscript are all cases performed by the senior author.

## 3. Results

### 3.1. Protrusio Acetabuli Classification System

The classification system proposed in this paper describes protrusio acetabuli in a more categorical fashion than prior systems (Table 3, Figure 1). This is anticipated to minimize errors and inconsistencies that may arise with interobserver measurement discrepancies. Class I protrusio in this system is described as a femoral head that is medialized but does not violate the ilioischial line. Class II protrusio describes a femoral head that has violated the ilioischial line but remains contained within the iliopectineal line. In class III protrusio, the femoral head has protruded medial to the iliopectineal line. Class IV protrusio in this system approximates periprosthetic protrusio, similar to the American Academy of Orthopaedic Surgeons class IV acetabular bone loss with pelvic discontinuity [29]. Traumatic hip and pelvic injuries or failed total hip arthroplasty cases need to be classified and managed with a different approach, and as this is separate from native hip protrusio; similarly, diagnosis and management of class IV protrusio will not be discussed further in this paper.

### 3.2. Managing Protrusio Acetabuli

Recommendations have been made in the past for the management of protrusio acetabuli based on the degree of deformity severity. Ranawat and Zahn in 1986 proposed that for protrusion less than 5 mm with a strong medial wall bone graft was not indicated, whereas for protrusion greater than 5 mm and with a thin but intact medial wall bone graft should be used [23]. They recommended reconstruction with bone graft and additional fixation devices in cases where the medial wall was grossly deficient [23]. These general guidelines remain helpful to consider when performing THA in patients with protrusio

acetabuli, though some of the implants commonly used during that area, such as cemented cup constructs, have fallen out of favor in recent decades [30].

Another important consideration in total hip arthroplasty for protrusio is the restoration of the normal hip center of rotation [20,31]. Given that the femoral head and acetabulum protrude medially in this condition, the goal is to recreate a more lateralized hip center to allow for appropriate hip biomechanics and reduce the risk of impingement and stiffness [32]. Digital templating is helpful in these cases to assess the current center of rotation and to plan the new center of rotation as well as any estimated leg length correction [33]. Templating also allows the surgeon to identify and anticipate defects in the acetabular wall prior to hip arthroplasty procedure and can be of significant benefit when planning to perform what is anticipated to be a complex total hip arthroplasty. The use of fluoroscopic image guidance and/or navigation should also be considered to assist with cup positioning, as landmarks are frequently altered in protrusio and can complicate the process of determining the most ideal cup position [34,35].

**Table 3.** Rubin classification for grading of protrusio acetabuli. (Note: For all classes of protrusio, pre-operative digital templating and intra-operative use of fluoroscopy are strongly recommended. The direct anterior approach to the hip is also recommended as a strategy to facilitate obtaining imaging and proper positioning of components.)

| Class | Ilioischial Line | Iliopectineal Line | Recommended Treatment |
|-------|------------------|--------------------|-----------------------|
| I | Intact (femoral head medialized) | Intact | Implant Placement: Lateralization of acetabular component to match native rim. Standard component needed, no need for screw augmentation.<br>Bone graft: Not likely required. |
| II | Violated | Intact | Implant Placement: Lateralization of acetabular component to rim, consider use of larger or jumbo component. Screw fixation optional. Consider lateralized acetabular liner.<br>Bone Graft: Likely required medially with impaction grafting. |
| III | Violated | Violated | Implant Placement: Lateralization of acetabular component, consider use of jumbo, deep profile, revision, or multi-hole component. Screw fixation is strongly recommended in multiple planes to augment construct stability. Consider lateralized acetabular liner.<br>Bone Graft: Universally required. Autograft reamings and morselized cortical bone from native head is recommended, consider use of additional allograft bone to fill large defects. Powdered antibiotic can be added to graft mixture to reduce risk of prosthetic joint infection. |

Note: The position of the native femoral head is used as the key reference in grading the deformity. The position of the medial edge of the femoral head relative to the landmarks above is the guiding principle to determine the severity of the protrusio deformity.

**Class I:** Medialized Femoral Head, does not violate Ilio-Ischial Line
**Class II:** Femoral Head is Between Ilio-Ischial and Ilio-Pectineal Lines
**Class III:** Femoral Head is Medial to Ilio-Pectineal Line

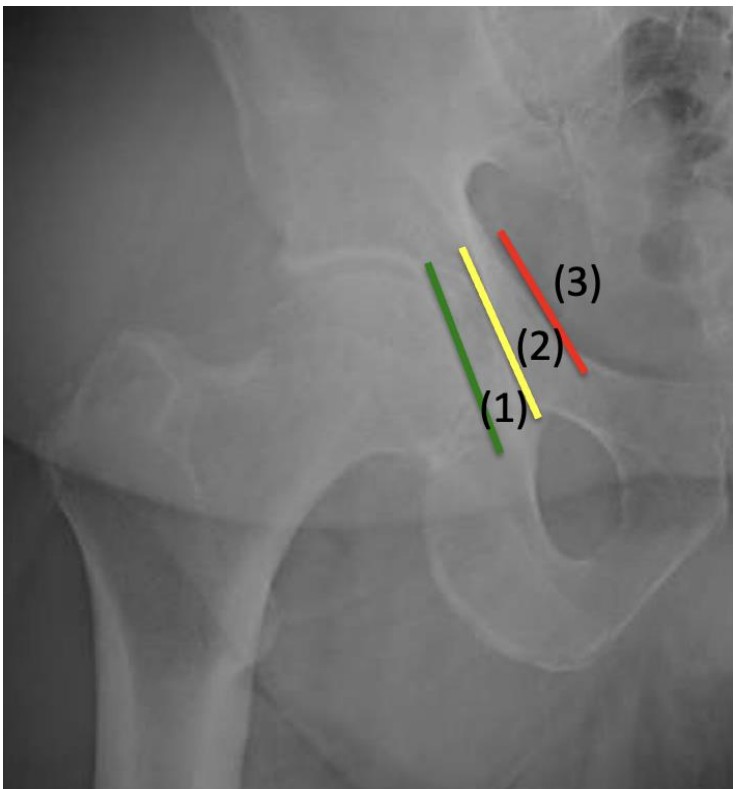

**Figure 1.** Pictorial representation of proposed protrusio classification.

Bone grafting options should be available when performing total hip arthroplasty in protrusio patients in order to restore deficient acetabular bone stock [32,36]. The native femoral head is frequently used as a source of bone graft during primary total hip arthroplasty [18], however, the surgeon should be prepared to use alternative graft sources in case the femoral head is for some reason unusable, such as if the femoral head is inadvertently contaminated during the procedure. Prior to the placement of the bone graft, the acetabulum should be prepared via careful reaming or scraping to stimulate bony bleeding and facilitate bone graft incorporation [18]. It should be kept in mind during the procedure that the goal is to lateralize the cup to the intact peripheral acetabular rim where adequate component fixation can be obtained, so sufficient graft should be made available to achieve this [18,32]. In cases where the cup appears to be seated more medially than desired despite the use of bone grafting, a lateralized acetabular liner can be used to further lateralize the center of rotation to a more satisfactory position [32].

*3.3. Protrusio Cases*

A few sample cases are described in this section to illustrate management options for different classes of protrusio based on this new classification system. The first case, depicted in Figure 2, is a patient with class I protrusio of the left hip. On the lateral XR of the patient's left hip (Figure 2b) the femoral head is seen to be medialized compared to the intact rim of the acetabulum, although the femoral head remains contained within the ilioischial line on the AP pelvis XR (Figure 2a). Digital templating was performed to determine the expected position of the components to be placed during the procedure as shown in

Figure 3. On the template, the center of the trunnion is seen medial to the center of the acetabular component, visually representing the lateralization of the hip center of rotation that will occur with placement of the acetabular component in the templated position. Figure 4 depicts the post-operative XR for this patient, with satisfactory lateralization of the hip center achieved without need for bone grafting.

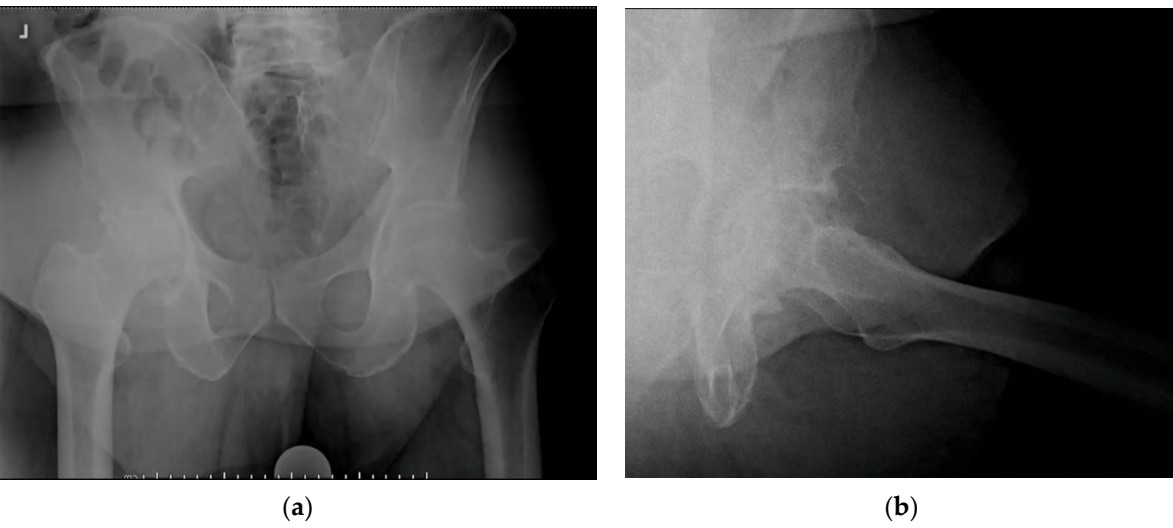

(**a**)                                                                              (**b**)

**Figure 2.** Class I protrusio patient ((**a**) AP pelvis XR; (**b**) lateral left hip XR).

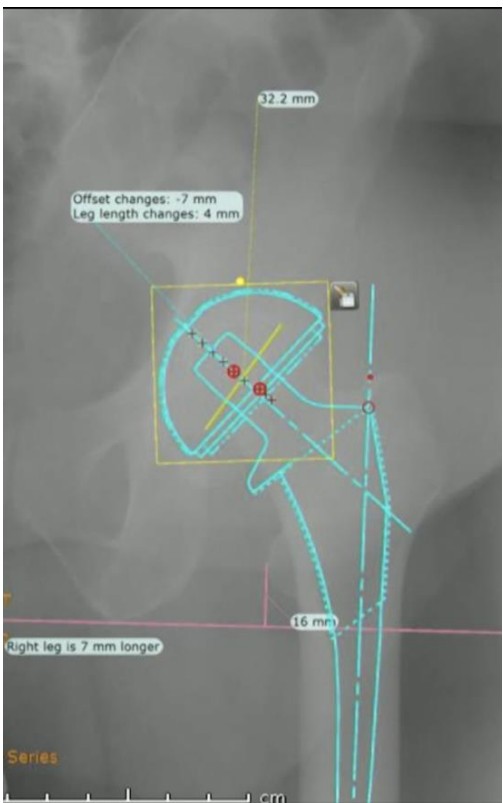

**Figure 3.** Class I protrusio patient template.

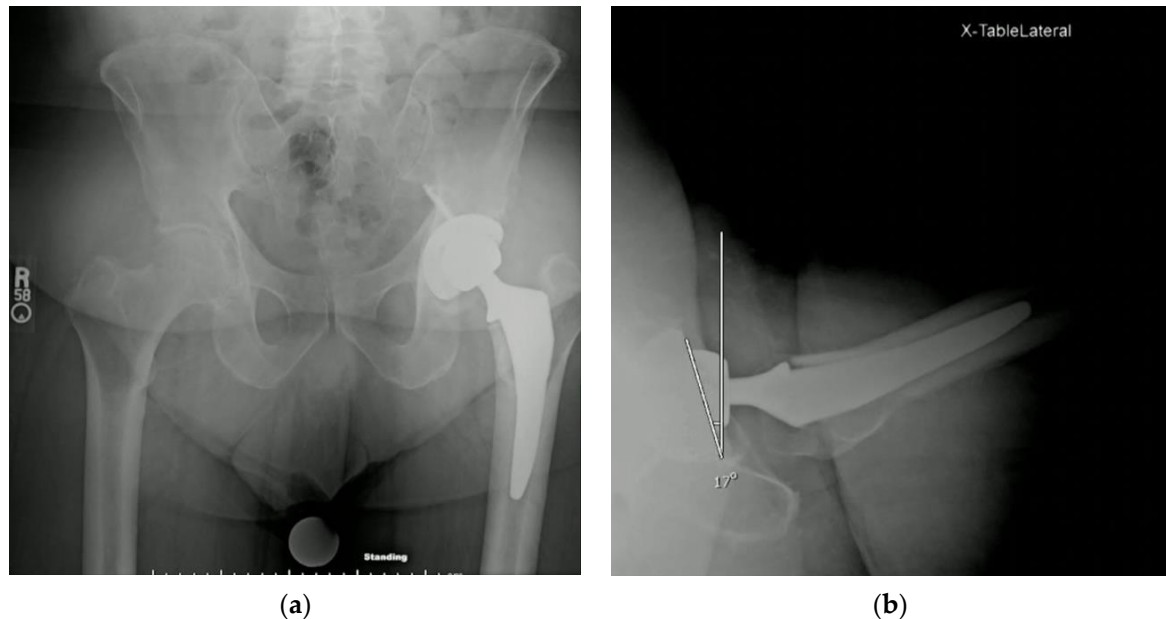

(**a**)  (**b**)

**Figure 4.** Class I protrusio patient post-operative XR ((**a**) AP pelvis XR; (**b**) lateral hip XR).

In Figure 5, the AP pelvis and lateral hip XRs are displayed for a patient with class III protrusio of the right hip and class II protrusio of the left hip. The patient's right femoral head appears to protrude medially past the ilioischial line on the AP pelvis XR (Figure 5a), while on the lateral XR of the right hip, it appears medial to the iliopectineal line as well (Figure 5b). Given that the patient's right hip had more severe protrusio and was more symptomatic, the right hip was selected for the patient to undergo total hip arthroplasty. Bone grafting was performed in this case using reamings from the femoral head to create morselized cancellous graft in addition to larger cortical fragments of the femoral head that were obtained by rongeur from the remaining shell of bone after reaming (Figure 6). The two sizes of bone graft were mixed together and used to fill the acetabular defect via progressive impaction grafting until the cup position could be lateralized to the acetabular rim. Powdered antibiotics (e.g., 1 g vancomycin) can be added to this bone graft mixture to decrease the risk of post-operative prosthetic joint infection.

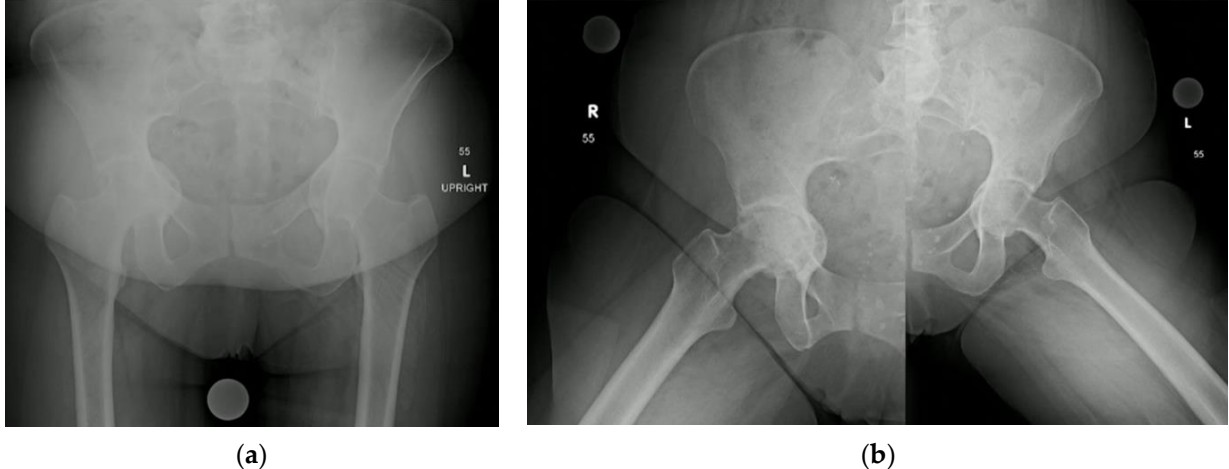

(**a**)  (**b**)

**Figure 5.** XRs of patient with class III protrusio of right hip and class II protrusio of left hip ((**a**) AP pelvis XR; (**b**) lateral hip XRs).

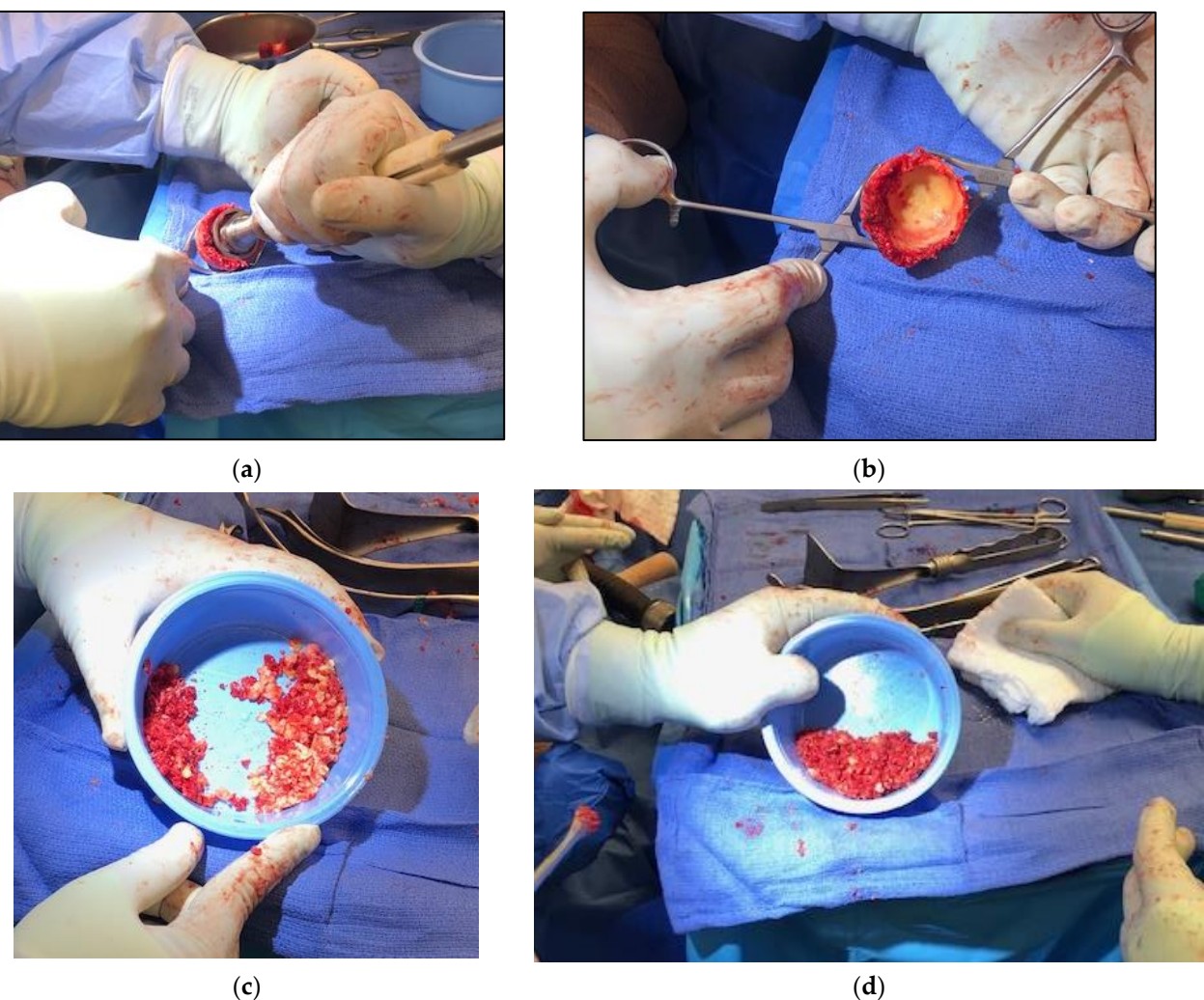

**Figure 6.** Collection and preparation of femoral head bone graft ((**a**) reaming of femoral head; (**b**) cortical shell of femoral head to be morselized; (**c**) two sizes of bone graft, finer graft from femoral head reamings on left and coarser graft from morselized femoral head cortex on right; (**d**) both sizes of bone graft mixed for impaction grafting).

The immediate post-operative XRs for this patient are depicted in Figure 7, and the 4-month and 15-month post-op XRs are shown in Figures 8 and 9, which document the remodeling of the protrusio defect back to a near-normal osteological appearance following the reconstructive procedure. This reconstitution of native bone stock is an essential goal for such a patient, in the event a future acetabular revision may be needed.

Other novel techniques have recently been described for the management of protrusio deformity, including in situ reaming of the hip with the femoral head in place after temporary stabilization of the head with K-wires to prevent femoral head spinning [37]. The acetabular cup is then placed into the reamed region. In protrusio cases where extraction of the femoral head is anticipated or found intra-operatively to be difficult, this method may provide an efficient way to proceed with total hip arthroplasty [37].

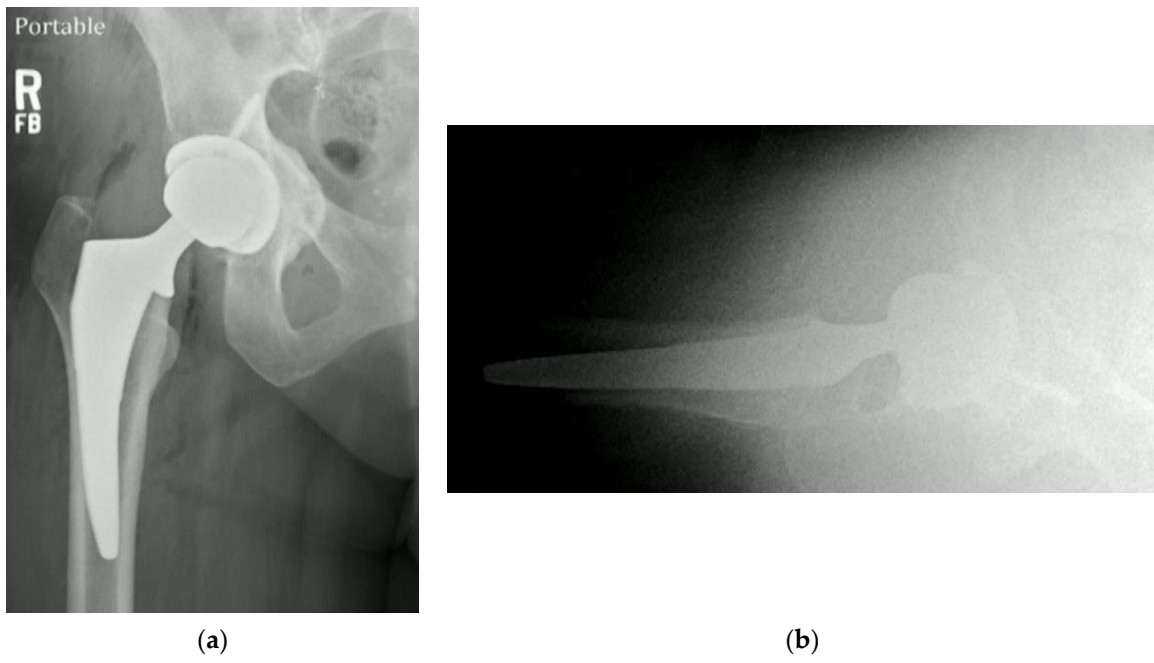

(**a**)                                                                 (**b**)

**Figure 7.** Class III right hip protrusio patient post-operative XR ((**a**) AP hip XR; (**b**) lateral hip XR).

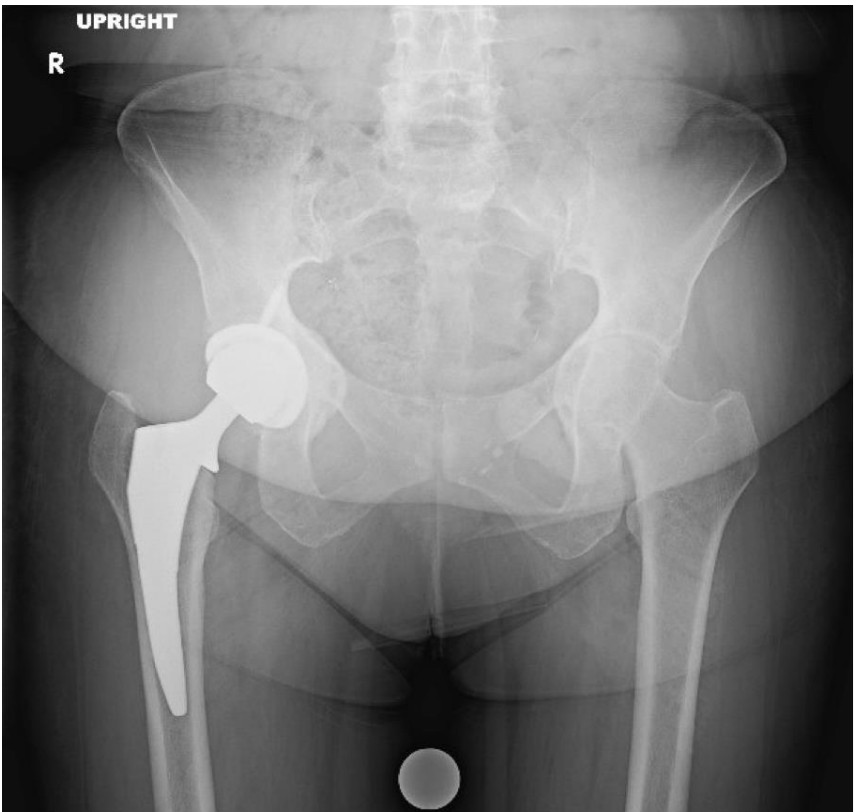

**Figure 8.** Four-month follow-up XR.

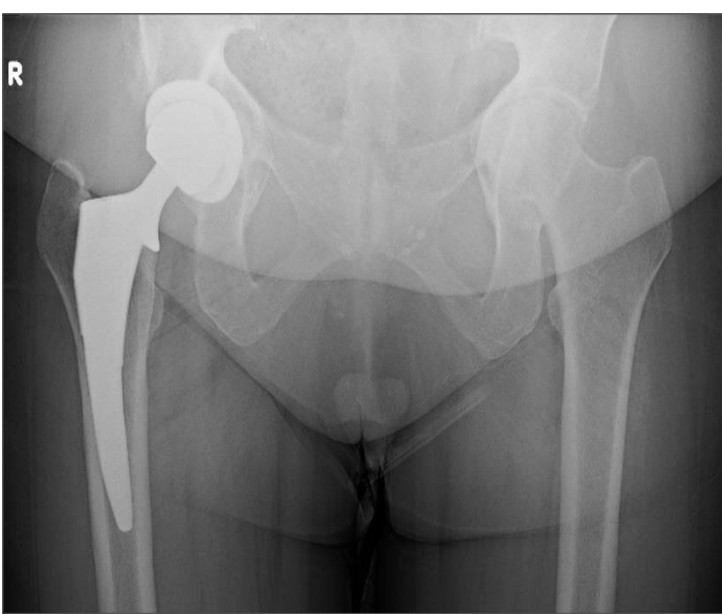

**Figure 9.** Fifteen-month follow-up XR.

*3.4. Technical Pearls*

Although protrusio acetabuli is a condition that can complicate the performance of total hip arthroplasty [19,38] with the appropriate preparation it is possible to achieve satisfactory outcomes [18]. There are several factors that, if addressed, can help make the procedure less complex. Pre-operative planning is an essential first step to any procedure and is of heightened importance in cases with a higher expected level of difficulty, such as total hip replacement in a protrusio patient [39]. This planning includes digital templating [33] and confirming that the instruments and bone graft materials that may be required will be present. The anesthesia plan should be determined as early as possible, and the use of general anesthesia should be considered if complete paralysis is anticipated to be needed to achieve the desired soft tissue tension and allow for adequate lateralization and distalization of components [40]. The use of traction and use of strategic femoral soft tissue releases can also help obtain sufficient exposure [41].

Fluoroscopy, robotics, and computer navigation are additional tools that can be of assistance when reconstructing the acetabulum [35]. These enabling technologies are essential to verify the anatomic correction of the hip center of rotation during the modern total hip arthroplasty procedure [34,35] and should be incorporated into the management strategy by default. Without verification of cup position during the implant procedure, it is not possible to reliably achieve the desired correction warranted by the protrusio classification system. The use of fluoroscopy is facilitated by performing these procedures with the patient supine, so utilization of the direct anterior approach to the hip should be strongly considered when planning for total hip arthroplasty in patients with protrusio [35]. It is the senior author's preference to use the direct anterior approach for complex total hip arthroplasty and for all protrusio cases, so as to visualize the osteological anatomy in real time via fluoroscopy during the hip reconstruction.

The surgeon should be prepared for difficult femoral head dislocation, which may necessitate the use of in situ osteotomies prior to dislocation [31]. Initial femoral neck resection and extraction help to facilitate subsequent extraction of the incarcerated and/or misshapen femoral head [31]. Performing comprehensive femoral releases as part of the initial exposure can allow for improved visualization of the acetabulum and increase the ease of acetabular instrumentation [41]. Typically, the senior author performs the posterolateral femoral release prior to initiating acetabular instrumentation during direct anterior approach total hip arthroplasty in Grades II and III protrusion cases in order to mobilize the contracted femur away from the acetabulum and improve the view of the

joint space. The principle of "Exposure before Instrumentation" is routinely emphasized, especially for the often-challenging surgical exposure needed for higher grade acetabular protrusion cases.

During direct anterior approach total hip arthroplasty, the Muëller, or double-footed retractor is an instrument that can be placed behind the acetabulum and used to depress and distract the femur in order to both improve exposure and protect the calcar. Either manual or table-assisted traction can also be applied during direct anterior approach total hip arthroplasty to further aid in the separation of the femur from the acetabular bone [41]. Small- and large-sized bone hooks are useful in allowing strategic application of force and distraction during these cases. In addition, the direct anterior approach allows the surgeon to place the patient supine, which facilitates the use of fluoroscopy during the procedure [35].

In class III protrusio, the hip may be extremely contracted, in which case a larger posterior sleeve release may be needed to achieve mobilization of the femur away from the acetabulum [41]. It should be remembered that void filling with autograft remains the gold standard for the management of acetabular defects in native hip protrusio [36]. Augments or other porous metal implants can be used strategically to fill large structural defects [42] and are then supplemented with autologous or allogeneic bone grafts, as may be warranted in revision total hip arthroplasty cases.

## 4. Conclusions

When thoughtfully approached, native hip acetabular protrusio is a condition that can be managed successfully via total hip arthroplasty with excellent patient outcomes. The use of the simplified, updated Rubin classification system such as the one proposed in this paper is anticipated to increase diagnostic consensus and therefore ease of communication when discussing protrusio acetabuli deformity, as well as to develop a standardized treatment strategy. There are several technical tools that have been described for total hip arthroplasty in protrusio acetabuli, which, if appropriately applied, may improve surgeon confidence and correction of the deformity during total hip arthroplasty in this patient population. As this is the first modern protrusio classification and management recommendation model described in over 35 years, further research with validation testing is warranted to determine consensus among a larger group of specialists in hip imaging and treatment, including radiologists and orthopedic surgeons.

**Author Contributions:** Conceptualization, L.E.R.; methodology, L.E.R. and L.A.; software/validation/ formal analysis, not applicable; investigation, L.E.R. and L.A.; resources, L.E.R., L.A. and Z.R.; data curation, L.E.R. and L.A.; writing–original draft preparation, L.A.; writing–review and editing, L.E.R., L.A. and Z.R.; visualization, L.E.R., L.A. and Z.R.; supervision, L.E.R.; project administration, L.E.R.; funding acquisition, L.E.R. All authors have read and agreed to the published version of the manuscript.

**Funding:** This research received no external funding.

**Institutional Review Board Statement:** Not applicable.

**Informed Consent Statement:** Not applicable.

**Data Availability Statement:** Not applicable.

**Conflicts of Interest:** The authors declare no conflict of interest.

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
