# Peer review of "Managing Native Hip Protrusio: Simplified Classification and Surgical Recommendations"

_2673-4036, doi:10.3390/osteology3010005_

Round 1

Reviewer 1 Report

Comments and Suggestions for Authors

The summarization of the main findings of the study:

Authors have done a work in that explains in a truly concise, orderly, and optimal way how to successfully manage acetabular protrusion, through an exposition about the different classification systems used in the past, as well as the explanation of other systems to be able to classify the bone loss that can be suffered in the Protusio Acetabulum.

They propose a novel classification system for the different cases of protrusion, in addition to finally using a series of "technical pearls" to conclude with the explanation of classification for total hip arthroplasty.

Detailed review report to the editor and authors:

Minor Comments:

Abstract: Should try to add some more information to the abstract so that it catches the reader's attention.

Line 120: Please put the information contained in the box in a table.

Line 223: Even though the technical pearls are authors' elaboration, all of them have a previous explanation that should be cited with a proper bibliography. Please, cite the paragraphs of the information presented.

Bibliography: Even if the information used in some paragraphs is from older sources, please try to update the bibliography with more current sources.

Author Response

Thank you very much for your review. Your summarization and the comments you made for improvement of the manuscript are greatly appreciated. Please find below responses to your comments. A revised version of the manuscript is also attached. 

Abstract: Additional information has been added. 

Line 120: I believe this line in the original submission was the beginning of the section entitled "Protrusio Acetabuli Classification System". This information is contained in Table 3 and depicted pictorially in Figure 1. Please let us know if you feel the table should be otherwise reformatted. 

Line 223: Citations have been added to the "Technical Pearls" section and included in the bibliography. 

Bibliography: The bibliography has been updated with more current sources. 

Reviewer 2 Report

Comments and Suggestions for Authors

The manuscript submitted for review is interesting, but needs some revisions before possible acceptance for publication. The title is far too long, please shorten it. Abstract-Please supplement with the mentioned general recommendations and technical pearls for total hip arthroplasty in the protrusio patient population. "Background" should be changed to "Introduction"-in this section and the next one, please introduce the correct citation of papers, according to the recommendations for authors of this journal. Make it clear in the manuscript what is the authors' own work and what are results taken from the literature. The division into traditional sections is missing: Material and Methods, Results, Discussion, which makes it not entirely clear what is the actual work of the authors of the manuscript.

Regards

Author Response

Thank you very much for your review. A revised version of the manuscript taking into account your comments has been attached below. Specifically, the title has been shortened; the abstract has been supplemented as advised with general recommendations and technical pearls for total hip arthroplasty in the protrusio patient population; the "Background" section has been changed to "Introduction", and the manuscript has been divided into traditional sections including "Materials and Methods", "Results", and "Discussion"; and citations of the literature have been more clearly documented.  

Round 2

Reviewer 2 Report

Comments and Suggestions for Authors

The authors have incorporated most of the recommended amendments.  However, the literature citation throughout the manuscript has not been corrected, I recommend that it be corrected according to the requirements for authors in this journal: for example, [2,3] ect. I have no other comments.

Regards

Author Response

Thank you again for your review. The format of the literature citations throughout the manuscript has been changed in accordance with the recommendations. An updated version of the manuscript is uploaded.
